# Effect of Perforation Dyeing Technique on Color Difference, Colorfastness, and Basic Density of Living Red-Heart Chinese Fir

Yiying Wang [1], Ruru Qu [1], Xiangwen Deng [1,2,3,*], Zhihong Huang [1,2,3], Wenhua Xiang [1,2,3] and Shuai Ouyang [1,2,3]

1   Faculty of Life Science and Technology, Central South University of Forestry and Technology, Changsha 410004, China; WangYYECO@163.com (Y.W.); qrr1109@163.com (R.Q.); T20101382@csuft.edu.cn (Z.H.); xiangwh2005@163.com (W.X.); yangshuai8613@613.com (S.O.)
2   National Engineering Laboratory for Applied Technology of Forestry & Ecology in South China, Changsha 410004, China
3   Huitong National Field Station for Scientific Observation and Research of Chinese Fir Plantation Ecosystem in Hunan Province, Huitong 438107, China
*   Correspondence: dengxw@csuft.edu.cn; Tel.: +86-731-85623458

**Abstract:** Red-heart Chinese fir is an excellent geographic provenance of *Cunninghamia lanceolata*, with high-value red heartwood. However, the formation of red heartwood is usually slow. To quickly cultivate red-heart Chinese fir, we studied perforation dyeing technology on living trees that were 7 years old and efficient in high-value red heartwood formation. Reactive dye (%), penetrant (%), $KH_2PO_4$ (%), and pH were selected as influencing factors, and an orthogonal test ($L_9(3)^4$) was used. The results showed that the total color difference between the experimental and CK groups ranged from 13.74 to 26.86 NBS, which was a significant visual perception (above 12 NBS). The total color difference before and after soaking in water for 6 h ranged from 2.30 to 5.12 NBS, which belonged to the detectable and identifiable value of the human eye (2~5 NBS). After the injection of the dye liquid, the wood basic density (WBD) was significantly affected after one year. After a comprehensive analysis of wood color difference, colorfastness, and WBD of the orthogonal test, the best dyeing process of juvenile red-heart Chinese fir was reactive dye: 0.8%, penetrant: 0.05%, $KH_2PO_4$: 0.3%, and pH: 3.5. The results of this study can provide a reference to improve the value of red-heart Chinese fir, a fast-cultivated, high-value decorative wood material.

**Keywords:** dyeing technology of living trees; fuzzy comprehensive evaluation; reactive dye; red-heart Chinese fir; wood basic density



## 1. Introduction

Wood is one of the most widely used of the four major materials (steel, cement, wood, and plastic), and its annual consumption is equivalent to the sum of the other three materials [1–3]. With the development of the economy and the increase in population, the market demand for wood products is also increasing [4]. The cultivation of artificial forest wood can effectively alleviate this problem [5]. Chinese fir (*Cunninghamia lanceolata* (Lamb.) Hook) is one of the most commercially important coniferous species, and is primarily distributed in southern China [6]. This conifer has produced many cultivars in large long-term production experiments. The wood of red-heart Chinese fir is also considered a high-value decorative material [7]. The basic density and fiber crystallization degree of red-heart fir are both greater than those of ordinary fir [8,9], and the price of its wood is also approximately 20% higher than that of ordinary fir [10], so red-heart Chinese fir is deeply loved by farmers and consumers. However, the resources of red-heart Chinese fir are limited and cannot meet the demand for high-value decorative wood materials [11]. To increase the resources of red-heart Chinese fir and cultivate high-value decorative wood

materials, artificial promotion of heartwood formation has become one of the methods of directional cultivation. Through wood dyeing technology, the yield of red-heart Chinese fir can be improved. Dyeing technology will be applied to simulate ordinary or low-quality wood to precious wood [12–14]. Through dyeing technology, the natural texture of the wood is highlighted and emphasized, visual aesthetic characteristics are displayed [15], inherent defects of wood are masked, surface properties of wood are modified, and the apparent value of wood itself is improved [16].

There are many methods of wood dyeing; according to the treatment form, they can be divided into scraps dyeing, single-plate dyeing, solid wood dyeing, standing wood dyeing, and so on [17]. Scraps dyeing generally refers to the dyeing treatment of broken wood pieces. Single-plate dyeing is a variety of thin wood or rotary plate dyeing processing methods, and the thickness is generally 0.2 to 1.0 mm. The dyed single plate can be used for the production of plywood, furniture veneer and composite solid wood floorboard, and decorative materials. Solid wood staining is the other wood or log staining. Compared with the previous wood forms, the thickness of this wood is large, and it is difficult to stain it evenly under conventional conditions. Therefore, generally in high-temperature, vacuum, high-pressure, and other conditions, the use of impregnated or cooking dyeing methods for dyeing material are mainly used for furniture production and interior decoration or cut thin wood. Standing wood dyeing is used to inject the dye solution from the base of the trunk into the growing living tree, or just-cut trees that still have vitality, so that the dye solution is transported to various parts through the capillary of wood, which is suitable for small wood with more sapwood [18,19]. On the other hand, living-tree perforation dyeing technology has the characteristics of simple equipment, convenient operation, no pollution to the environment, and energy savings.

Research on the dyeing technology of living trees first began in Japan, which first used fluid flow to conduct dyeing experiments on trees [20]. In 1999, Zhao [21] applied for the patent "Method of Vertical Wood Dyeing" and successfully realized the dyeing of single-color dyeing and red-blue multicolor dyeing. In 2008, Chen et al. [22] first analyzed and obtained the optimal technological ratio of the two dyes, which laid the foundation for living wood dyeing. In 2000, Xiong et al. [23] adopted reactive (KN-3G) dye, and found that among the four factors of reactive dye %, pH, dyeing time, and penetrant %, the pH had the greatest influence on the dyeing rate and color difference. With the continuous development and maturity of dyeing technology, scholars have explored the best effects of dyeing, and have paid more attention to exploring the influence of dyeing-solution injection on wood growth and physiological activities. In 2017, when Cao et al. [24] conducted live wood dyeing, they found that after the dye fluid was injected into the tree, the maximum liquid flow rate, net photosynthetic rate, air penetrant, and transpiration rate of the trunk were significantly lower than those of the CK group, and the dye liquid conducted longitudinally in both directions along the trunk. Subsequently, Fan et al. [25,26]. analyzed in detail the changes in photosynthetic physiology and trunk fluid flow after the introduction of dye fluid, and discussed the relationship between these physiological changes and the amount of dye.

Although some achievements have been made in the perforation dyeing technology of living trees, the results have been mainly concentrated on broad-leaved small-diameter trees. Although the growth status and physiological activity of living wood have been studied, long-term observations of the influence of dyeing solution on the growth of living trees are lacking. In addition, it can be found from the above studies that the main factors affecting the dyeing of living trees include the type of tree, concentration, pH value, and the type of test materials. The main purposes of this study were: (1) to compare the color difference and colorfastness before and after the different dyeing treatments; (2) to determine the effect of different treatments on wood basic density after 1 year of dyeing; and (3) to determine the best living tree dyeing technology on red-heart Chinese fir for the color difference, colorfastness, and basic density using a comprehensive evaluation method.

## 2. Materials and Methods

### 2.1. Site Description

The study was conducted at Chenshan Forestry Farm (27°04′-27°36′ N, 114°00′–114°47′ E; 110–1300 m ASL) in Anfu County, Jiangxi Province. The study area is characterized by a subtropical humid climate, with an average annual temperature of 17.7 °C (the average January and July temperatures are 5.9 and 28.9 °C, respectively), average annual precipitation of 1663 mm, average sunshine duration of 1649 h, and an average annual frost-free period of 279 d (based on data from 1970 to 2018). The pH value of the surface soil (0–10 cm) is approximately 4.27 [27]. A 7-year-old juvenile red-heart Chinese fir in Chenshan Forestry Farm, Jiangxi Province was selected as the sample. Chenshan Forestry Farm is well known for its resources of red-heart Chinese fir. The understory consisted mainly of *Sapium sebiferum*, *Vernicia fordii*, *Phyllostachys edulis*, and *Choerospondias axillaris* [28].

### 2.2. Dye Selection

Synthetic dyes commonly used in wood dyeing include direct dyes, acid dyes, basic dyes, and reactive dyes [29]. We added reactive dyes to the dyes because the reactive dyes have better water resistance and light resistance, which is beneficial to living wood dyeing [30,31].

To improve the dyeing effect, we needed to mix some additives with the dyes. Dyeing auxiliaries are commonly used according to their uses, including as a permeating agent, promoting agent, leveling agent, and fixing agent [32]. The effect of penetrant on living wood dyeing is more significant, so it was added to the dyes.

The study area is at a low latitude and lacks available phosphorus [33]. Soil phosphorus is an important factor that restricts the rapid growth and high yield of artificial forests in southern China, and even affects the balance and stability of ecosystems [34]. Therefore, we added different concentrations of $KH_2PO_4$ to the dyes.

The pH of a dyeing solution will affect the time and color difference of wood dyeing [35]. The pH of active dyes is generally between 9 and 10 [36]. When the pH value of the dye solution is small, the wood dyeing time is short, the dye is evenly distributed and stable in the wood, and the color difference before and after dyeing is small. When the pH of the dyeing solution is high, the dyeing time is relatively long, and the color difference of wood before and after dyeing is large, but the colorfastness is unstable.

The mass fraction, permeability, and pH value of reactive dyes were taken as the influencing factors and treated by orthogonal experiments, which are shown in Table 1. We established the $L_9(3)^4$ orthogonal experimental, which included 4 influencing factors all with 3 concentration levels. In the experiment, there were 9 treatments with 3 replicates per treatment, and the control group used distilled water. According to the orthogonal table, the experimental treatment could be distributed evenly, and the number of experiments could be reduced [37]. The best dyeing process combination for red-heart Chinese fir live dyeing was confirmed by orthogonal testing.

**Table 1.** L$_9$(3)$^4$ orthogonal experimental table.

| Process Number | Reactive Dye (%) | Penetrant (%) | KH$_2$PO$_4$ (%) | pH |
|:---:|:---:|:---:|:---:|:---:|
| 1 | 0.2 | 0.01 | 0.1 | 3.5 |
| 2 | 0.2 | 0.05 | 0.2 | 4.5 |
| 3 | 0.2 | 0.10 | 0.3 | 5.5 |
| 4 | 0.5 | 0.01 | 0.2 | 5.5 |
| 5 | 0.5 | 0.05 | 0.3 | 3.5 |
| 6 | 0.5 | 0.10 | 0.1 | 4.5 |
| 7 | 0.8 | 0.01 | 0.3 | 4.5 |
| 8 | 0.8 | 0.05 | 0.1 | 5.5 |
| 9 | 0.8 | 0.10 | 0.2 | 3.5 |
| CK | 0 | 0 | 0 | 0 |

The three levels of the four influencing factors were used in the orthogonal test L$_9$(3)$^4$ with 3 replicates per treatment, and CK (the control group) used distilled water. The three levels of reactive dye were 0.2%, 0.5%, and 0.8%. The three levels of penetrant were 0.01%, 0.05%, and 0.10%. The three levels of KH$_2$PO$_4$ were 0.1%, 0.2%, and 0.3%, and the three levels of pH were 3.5, 4.5, and 5.5.

### 2.3. Dyeing Method

Three replicates (i.e., 3 trees) were set for each treatment in the orthogonal test table; each tree was equipped with 2000 mL of dye fluid in the infusion bag, and three blanks (distilled water) controls were set. Before dye injection, each of 300 Chinese fir trees in the sample plot was measured first, and 30 Chinese fir trees with the same age, healthy growth, and terrain conditions were selected. When injecting, we drilled a hole in the north and south direction of the trunk 15 cm away from the ground, avoiding the damaged parts of the trunk. The hole diameter was about 0.5 cm, and the hole was deep into the middle of the trunk. We installed the dyeing infusion device, and the gap between the orifice and the infusion needle was sealed with tree callus to prevent the entry of bacteria. After one year of the experiment, the three indexes of the color difference, colorfastness, and basic density were determined by cutting the sample wood.

### 2.4. Measurement Indexes and Experimental Methods

2.4.1. Color Difference Measurement

The color was determined using the CIE1976 (*L**, *a**, *b**) surface color system [38]. The brightness index (*L**), red-green axis color index (*a**), and yellow-blue axis color index (*b**) of the dyed red-heart Chinese fir wood were measured with a color spectrum colorimeter (CS-200). The color difference was determined by the average value of $\Delta L^*$, $\Delta a^*$, and $\Delta b^*$ (Formulas (1)-(3)). The total color difference ($\Delta E^*$) was calculated between the control group and different dyeing treatments after perforation dyeing for 1 year (Formula (4)). Color difference units used were NBS units, which were established by the National Bureau of Standards. When the color difference $\Delta E^* = 1$, it is called one NBS unit [39].

$$\Delta L^* = L_1{}^* - L_0{}^* \tag{1}$$

$$\Delta a^* = a_1{}^* - a_0{}^* \tag{2}$$

$$\Delta b^* = b_1{}^* - b_0{}^* \tag{3}$$

$$\Delta E^* = \sqrt{(\Delta L^*)^2 + (\Delta a^*)^2 + (\Delta b^*)^2} \tag{4}$$

where $\Delta L^*$ is the brightness difference (a greater positive value means brighter, and a greater negative value means darker); $\Delta a^*$ is the red-green axis color difference index (the larger the positive value, the deeper red; and the larger the negative value, the deeper green); $\Delta b^*$ is the yellow-blue axis color difference index (the larger the positive value, the more yellow; and the larger the negative value, the deeper the blue); and $\Delta E^*$ is the total color difference (the larger the value, the greater the color change).

### 2.4.2. Colorfastness and Wood Basic Density (WBD) Measurement

Small pieces of wood boards with dimensions of 5 cm (length) × 4 cm (width) × 2 cm (thickness) were sawed to determine the colorfastness. Each wood board was soaked in water for 6 h. Three points were selected on each wood board to measure the $L^*$, $a^*$, and $b^*$. The colorfastness was determined by the color difference of the average value of $\Delta L^*$, $\Delta a^*$, and $\Delta b^*$ before and after soaking in water.

Wood regular density was divided into four kinds, and the wood basic density (WBD) and wood air-dry density are commonly used. WBD is one of the most crucial wood properties in trees, and mainly determines the end use of wood for the industry. Wood air-dry density is suitable for measuring the density of wood in dry air for a long time [40]. The test object was Chinese fir in the natural environment, so it was more suitable for measuring the basic density. The WBD of samples was determined by following per under GB/T 1933–2009 [41], and 3 replicates were taken for each group of treated samples.

### 2.5. Data Analysis

The combined entropy value method and fuzzy comprehensive evaluation method were carried out in the R package 4.0.3 [42], which was used to comprehensively evaluate the best process of the perforation dyeing technique on living red-heart Chinese fir.

### 2.5.1. Establishment of the Evaluation Set

Three indexes of color difference, colorfastness, and basic density of dyed Chinese fir were obtained by orthogonal testing, and the evaluation matrix of dyeing was established. The mean $q_{ik}$ (Formula (5)) of the corresponding index at each level of each factor was first calculated according to the literature [43]:

$$q_{ik} = \frac{\sum\limits_{k=1}^{3} x_{ij(k)}}{3} \tag{5}$$

where $i$ = 1, 2, 3, 4 indicates the influencing factors in orthogonal test, namely reactive dye %, penetrant %, $KH_2PO_4$, and the pH value; $j$ = 1, 2, 3, ... 9 indicates the number of orthogonal tests, for a total of 9 tests; $k$ = 1, 2, 3, ... 9 indicates the concentration level of each influence factor; and $x_{ij(k)}$ is the original value of an evaluation index at the $k$ level of the $i$ factor and $j$ processing number.

Since the stipulation of fuzzy mathematics, the evaluation indexes should be evaluated comprehensively according to the principle of "selecting the best", so the side-effect indexes in the evaluation indexes should be changed incrementally to make them meet the evaluation requirements. Among the three evaluation indexes selected in this study, colorfastness is a negative action index, which was treated incrementally as follows (Formula (6)):

$$q'_{ik} = \frac{1}{q_{ik}} \tag{6}$$

where $y$ after the increment transformation satisfies the increment condition, but is still a normal set. It is blurred to eliminate its dimension, and each element in the ordinary set is the weighted average (Formula (7)):

$$r_{ik} = \frac{q_{ik}}{\sum\limits_{k=1}^{3} q_{ik}} \tag{7}$$

### 2.5.2. Determination of Evaluation Indicators

In the comprehensive evaluation of multiple indexes, the main methods to determine the weight of indexes include the subjective weight method and the objective weight method [44]. Entropy is a measure of uncertainty [45]. In the comprehensive evaluation of

multiple indexes, the entropy value can be used as the basis to determine the weight of indexes, which makes up for the defect of not considering the data characteristics when using principal component analysis to determine the weight [46,47]. This study used the entropy method to determine the weight of the index.

2.5.3. Fuzzy Matrix Synthesis

Combining the dimensionless evaluation matrix R of each evaluation index under different influencing factors with the weight vector *A* of each evaluation index obtains the final evaluation result vector *B*. After normalizing the data in result vector *B* (Formula (8)), the optimal concentration of each influencing factor was selected:

$$B = A \times R = \begin{pmatrix} W_1 & W_2 & W_3 \end{pmatrix} \times \begin{pmatrix} r_{11} & r_{12} & r_{13} \\ r_{21} & r_{22} & r_{23} \\ r_{31} & r_{32} & r_{33} \end{pmatrix} = \begin{pmatrix} B_1 & B_2 & B_3 \end{pmatrix} \quad (8)$$

where $B_1$, $B_2$, and $B_3$ are the comprehensive evaluation values of different evaluation indexes under three different concentrations of a certain influencing factor, and the largest is the best concentration of this factor.

## 3. Results

### 3.1. Effects of Dyeing Factors on Color Difference

The color difference of red-heart Chinese fir wood before and after dyeing represented the total color difference ($\Delta E_1^*$). As shown in Figure 1, the total color difference ($\Delta E_1^*$) of the 9 groups was different. The better the dyeing effect was, the greater the color difference value. As shown in Table 2, the differences of the 9 experimental treatments were obtained by ANOVA, and the lightness difference ($\Delta L^*$) between the experimental and CK groups was negative, indicating that the lightness of the dyed wood decreased and the wood color darkened. The red-green axis color difference index ($\Delta a^*$) was positive. The yellow-blue axis color difference index ($\Delta b^*$) changed little, indicating that the wood color was a deeper red after dyeing. The total color difference ($\Delta E_1^*$) range between the experimental and CK groups was 13.74~26.86 NBS, which was a very significant visual perception (up to 12 NBS) [48]. NBS, the unit of color difference, was established by the National Bureau of Standards. When the color difference $\Delta E_1^* = 1$, it was called one NBS unit. The results showed that the liquid dyeing effect was better in the red-heart Chinese fir wood, and the wood color turned an obvious red.

**Table 2.** Color difference indexes of red-heart Chinese fir wood with different dyeing treatments.

| Process Number | L* | a* | b* | ΔL* | Δa* | Δb* | ΔE₁* |
|---|---|---|---|---|---|---|---|
| 1 | 71.07 ± 1.95 | 18.01 ± 2.89 | 14.32 ± 0.94 | −9.76 ± 1.74 | 12.17 ± 2.94 | −1.18 ± 1.03 | 15.81 ± 3.12 c |
| 2 | 72.49 ± 1.77 | 16.93 ± 3.30 | 12.02 ± 1.44 | −8.33 ± 1.79 | 11.10 ± 3.42 | −3.47 ± 1.55 | 14.51 ± 3.63 bc |
| 3 | 71.97 ± 3.80 | 15.53 ± 5.11 | 15.88 ± 1.82 | −8.85 ± 3.50 | 9.69 ± 5.15 | 0.39 ± 1.68 | 13.88 ± 5.50 abc |
| 4 | 73.57 ± 4.25 | 16.69 ± 3.80 | 12.17 ± 0.41 | −7.25 ± 4.19 | 10.86 ± 3.93 | −3.32 ± 0.65 | 13.74 ± 5.40 abc |
| 5 | 67.59 ± 4.88 | 22.82 ± 6.05 | 11.08 ± 0.63 | −13.23 ± 7.54 | 16.98 ± 6.00 | −4.42 ± 0.79 | 22.23 ± 6.94 abc |
| 6 | 69.67 ± 4.70 | 13.73 ± 6.04 | 18.67 ± 1.40 | −11.15 ± 4.48 | 7.90 ± 6.13 | 3.17 ± 1.33 | 14.70 ± 6.88 abc |
| 7 | 65.16 ± 2.83 | 23.71 ± 4.12 | 12.29 ± 0.76 | −15.66 ± 3.12 | 17.87 ± 4.30 | −3.20 ± 2.23 | 24.09 ± 5.21 abc |
| 8 | 65.24 ± 1.83 | 26.94 ± 2.88 | 11.45 ± 0.54 | −15.59 ± 1.11 | 21.11 ± 2.80 | −4.05 ± 0.49 | 26.66 ± 2.80 ab |
| 9 | 63.17 ± 5.65 | 25.43 ± 6.33 | 11.04 ± 2.11 | −17.65 ± 5.44 | 19.59 ± 6.42 | −4.46 ± 2.22 | 26.86 ± 8.54 a |

Note: *L**, brightness index of dyed red-heart Chinese fir wood; *a**, color indexes at the red-green axis; *b**, color indexes at the yellow-blue axis; $\Delta L^*$, lightness difference; $\Delta a^*$, red-green axis color difference index; $\Delta b^*$, yellow–blue axis color difference index; $\Delta E_1^*$, the total color difference of red-heart Chinese fir wood before and after dyeing; ±, standard deviation; c, bc, abc, ab, a, results of ANOVA.

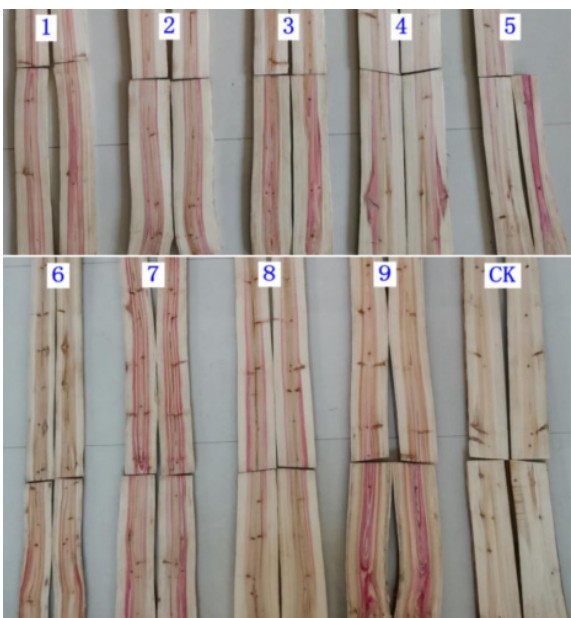

**Figure 1.** Dyed Chinese fir under different dyeing treatments. Note: After dyeing treatment, the color of wood in the experimental group 1-9 was significantly redder than that in CK.

Through extreme differential analysis of each color difference index (Table 3), we found that the four factors had the following effects on the total color difference ($\Delta E_1^*$) after dyeing: reactive dye % ($A$) > pH value ($D$) > penetrant % ($B$) > $KH_2PO_4$% ($C$).

As shown in Figure 2, the change in the total color difference ($\Delta E_1^*$) was not due to a single cause, but the reactive dye %, penetrant %, $KH_2PO_4$%, and pH factors worked together to produce interactions. Among them, the reactive dye % had significant effects on the brightness index difference ($\Delta L^*$), ($\Delta a^*$), ($\Delta b^*$), and total color difference ($\Delta E_1^*$). The change in concentration had the greatest influence on the total color difference ($\Delta E_1^*$). The total color difference ($\Delta E_1^*$) of dyed Chinese fir was used as the evaluation index. The best dyeing process for Chinese fir live dyeing was $A_3B_2C_3D_1$, which meant the best formula was: mass fraction of reactive dye: 0.8%; mass fraction of penetrant: 0.05%; mass fraction of $KH_2PO_4$: 0.3%; and pH: 3.5.

**Table 3.** Range analysis of the color difference indexes of dyed red-heart Chinese fir wood.

| Four Factors | | Levels | $\Delta L^*$ | $\Delta a^*$ | $\Delta b^*$ | $\Delta E_1^*$ |
|---|---|---|---|---|---|---|
| Code | Factors | | | | | |
| $A$ | Reactive dye (%) | 0.2% | −8.98 | 10.99 | −1.42 | 14.73 |
| | | 0.5% | −10.54 | 11.91 | −1.52 | 16.89 |
| | | 0.8% | −16.30 | 19.52 | −3.90 | 25.87 |
| | | $R$ | 7.32 | 8.53 | 2.48 | 11.14 |
| $B$ | Penetrant (%) | 0.01% | −10.89 | 13.63 | −2.57 | 17.88 |
| | | 0.05% | −12.38 | 16.39 | −3.75 | 21.13 |
| | | 0.10% | −12.55 | 12.39 | −2.41 | 18.48 |
| | | $R$ | 1.66 | 4.00 | 1.34 | 3.25 |
| $C$ | $KH_2PO_4$ (%) | 0.1% | −12.16 | 13.73 | −0.68 | 19.06 |
| | | 0.2% | −11.08 | 13.85 | −3.75 | 18.37 |
| | | 0.3% | −12.58 | 14.85 | −2.41 | 20.06 |
| | | $R$ | 1.50 | 1.12 | 3.07 | 1.70 |
| $D$ | pH | 3.5 | −13.55 | 16.25 | −3.35 | 21.64 |
| | | 4.5 | −11.71 | 12.29 | −1.17 | 17.76 |
| | | 5.5 | −10.56 | 13.89 | −2.33 | 18.09 |
| | | $R$ | 2.98 | 3.96 | 2.18 | 3.87 |

**Table 3.** *Cont.*

| Four Factors | | Levels | ΔL* | Δa* | Δb* | ΔE₁* |
|---|---|---|---|---|---|---|
| Code | Factors | | | | | |
| | Sort Factors | | $A > D > B > C$ | $A > B > D > C$ | $C > A > D > B$ | $A > D > B > C$ |

Note: ΔL*, lightness difference; Δa*, red–green axis color difference index; Δb*, yellow–blue axis color difference index; ΔE₁*, the color difference of red-heart Chinese fir wood before and after dyeing; R, range.

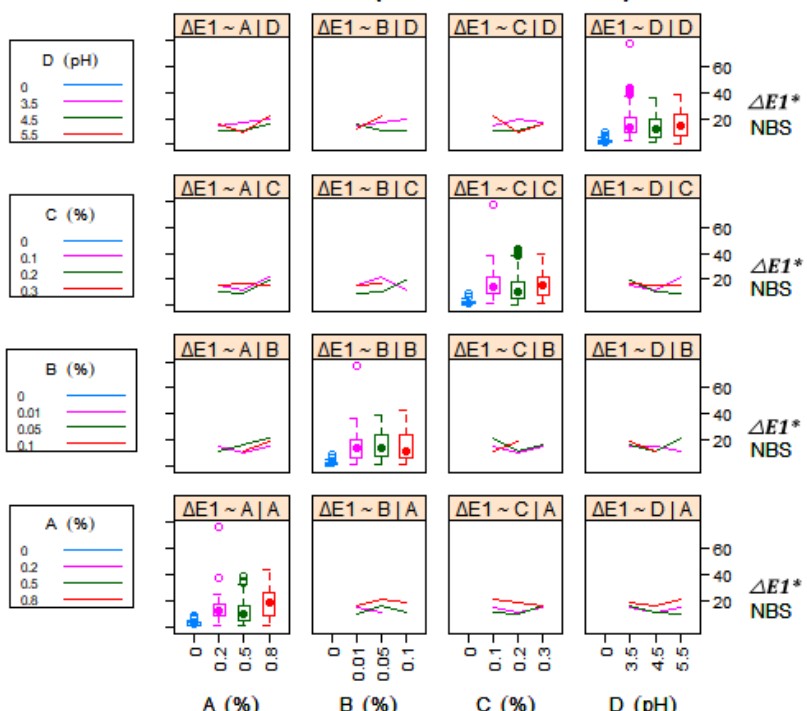

**Figure 2.** Effects of four factors (all with three levels) on $\Delta E_1^*$. $\Delta E_1^*$ is the total color difference (NBS). NBS is the unit of color difference, established by the National Bureau of Standards. When the color difference $\Delta E_1^* = 1$, it is called one NBS unit. A is reactive dye (%) with three levels of 0.2% ($A_1$), 0.5% ($A_2$), and 0.8% ($A_3$). B is penetrant (%) with three levels of 0.01% ($B_1$), 0.05% ($B_2$), and 0.10% ($B_3$). C is $KH_2PO_4$ (%) with three levels of 0.1% ($C_1$), 0.2% ($C_2$), and 0.3% ($C_3$). D is pH with three levels of 3.5 ($D_1$), 4.5 ($D_2$), and 5.5 ($D_3$). The $y \sim AIA$, $y \sim BIB$, $y \sim CIC$, and $y \sim DID$ represent the leading factors in the interaction. $A_3B_2C_3D_1$ was the best formula.

### 3.2. Effect of the Dyeing Factors on the Colorfastness

The color difference of red-heart Chinese fir-dyed wood before and after soaking in water indicated its colorfastness ($\Delta E_2^*$). The smaller the color difference was, the fewer dyes precipitated from the wood during water immersion and the higher the water colorfastness of the dyeing material. As shown in Table 4, the differences of the nine experimental treatments were obtained by ANOVA; and the brightness difference ($\Delta L^*$) was positive, and the red-green axis color difference index ($\Delta a^*$) was negative and decreased with increasing reactive dye %. The yellow-blue axis color difference index ($\Delta b^*$) was not obvious, indicating that the dyeing material became bright, red became lighter, and the color brightness decreased. The colorfastness difference ($\Delta E_2^*$) of wood before and after soaking in water for 6 h ranged from 2.30 to 5.12 NBS, which belonged to the detectable and identifiable value of the human eye (2~5 NBS).

**Table 4.** Color change of dyed red-heart Chinese fir wood before and after soaking in water for 6 h.

| Treatment Number | Before Soaking | | | After Soaking | | | Color Parameter Difference | | | Total Color Difference |
|---|---|---|---|---|---|---|---|---|---|---|
| | $L^*$ | $a^*$ | $b^*$ | $L^*$ | $a^*$ | $b^*$ | $\Delta L^*$ | $\Delta a^*$ | $\Delta b^*$ | $\Delta E_2^*$ |
| 1 | 70.07 ± 2.32 | 20.36 ± 2.47 | 13.13 ± 1.84 | 71.00 ± 2.40 | 18.51 ± 2.59 | 13.09 ± 1.79 | 0.92 ± 1.34 | −1.84 ± 1.56 | −0.04 ± 0.66 | 2.61 ± 1.50 c |
| 2 | 71.77 ± 3.02 | 16.49 ± 1.79 | 16.24 ± 1.78 | 72.05 ± 2.55 | 15.06 ± 2.38 | 15.73 ± 1.77 | 0.28 ± 1.39 | −1.43 ± 0.89 | −0.51 ± 1.34 | 2.30 ± 1.26 bc |
| 3 | 67.52 ± 2.01 | 18.64 ± 2.35 | 16.08 ± 2.55 | 68.12 ± 3.01 | 16.78 ± 2.78 | 16.43 ± 1.90 | 0.61 ± 1.46 | −1.87 ± 0.95 | 0.35 ± 1.26 | 2.72 ± 1.09 abc |
| 4 | 73.57 ± 5.89 | 16.83 ± 4.98 | 13.64 ± 0.93 | 74.57 ± 5.09 | 14.53 ± 4.00 | 13.74 ± 1.03 | 0.99 ± 0.99 | −2.30 ± 1.14 | 0.10 ± 0.43 | 2.63 ± 1.41 abc |
| 5 | 65.11 ± 4.78 | 24.64 ± 5.97 | 12.40 ± 1.02 | 67.01 ± 4.36 | 22.62 ± 5.57 | 13.82 ± 1.51 | 1.91 ± 1.88 | −2.02 ± 1.51 | 1.42 ± 1.07 | 3.53 ± 1.81 abc |
| 6 | 60.60 ± 1.20 | 20.22 ± 0.88 | 13.81 ± 2.22 | 62.41 ± 1.65 | 18.30 ± 0.72 | 15.77 ± 2.18 | 1.81 ± 2.18 | −1.93 ± 1.18 | 1.96 ± 1.95 | 3.84 ± 2.47 abc |
| 7 | 61.93 ± 2.79 | 25.18 ± 3.50 | 15.63 ± 1.88 | 65.12 ± 3.95 | 22.13 ± 3.67 | 15.59 ± 2.00 | 3.18 ± 1.69 | −3.05 ± 1.26 | −0.03 ± 2.44 | 5.12 ± 1.64 abc |
| 8 | 64.36 ± 2.62 | 26.56 ± 4.98 | 13.55 ± 1.75 | 65.90 ± 3.24 | 23.81 ± 4.24 | 13.90 ± 0.81 | 1.54 ± 2.11 | −2.74 ± 2.01 | 0.35 ± 1.87 | 4.22 ± 2.06 ab |
| 9 | 62.28 ± 6.53 | 24.42 ± 6.15 | 12.75 ± 1.71 | 65.35 ± 5.18 | 22.71 ± 5.75 | 13.39 ± 1.15 | 3.08 ± 1.80 | −1.71 ± 0.95 | 0.64 ± 1.90 | 4.01 ± 2.12 c |

Note: $L^*$, brightness index of dyed red-heart Chinese fir wood; $a^*$, color indexes at the red-green axis; $b^*$, color indexes at the yellow-blue axis; $\Delta L^*$, lightness difference; $\Delta a^*$, red–green axis color difference index; $\Delta b^*$, yellow–blue axis color difference index; $\Delta E_2^*$, total color difference of red-heart Chinese fir dyed wood before and after soaking in water for 6 h; ±, standard deviation; abc, results of ANOVA.

According to the range analyzed of the colorfastness difference ($\Delta E_2^*$) before and after soaking in water (Table 5), we found that the four factors had the following effects on the total colorfastness of dyed red-heart Chinese fir: reactive dye % ($A$) > $KH_2PO_4$% ($C$) > pH value ($D$)> penetrant % ($B$). The colorfastness difference ($\Delta E_2^*$) of dyed red-heart Chinese fir was used as the evaluation index. The best dyeing process for Chinese fir live dyeing was $A_1B_2C_2D_3$. This meant that the best formula was as follows: mass fraction of reactive dye of 0.2%, mass fraction of penetrant of 0.05%, mass fraction of $KH_2PO_4$ of 0.2%, and pH value of the dyeing solution of 5.5.

The variance analysis (Table 6) that was conducted for the color difference ($\Delta E_2^*$) of the dyed material before and after soaking in water found that the reactive dye % had a very significant effect on the colorfastness of Chinese fir wood ($p < 0.001$), while the penetrant%, $KH_2PO_4$%, and the pH value did not affect the colorfastness of dyed red-heart Chinese fir wood ($p > 0.05$).

**Table 5.** Range analysis of color difference of dyed red-heart Chinese fir wood before and after soaking in water for 6 h.

| Horizontal Analysis | Factors | | | |
|---|---|---|---|---|
| | $A$ | $B$ | $C$ | $D$ |
| $X_1$ | 2.55 | 3.45 | 3.56 | 3.38 |
| $X_2$ | 3.33 | 3.35 | 2.98 | 3.75 |
| $X_3$ | 4.45 | 3.52 | 3.79 | 3.19 |
| $R$ | 1.91 | 0.17 | 0.81 | 0.57 |
| Sort Factors | $A > C > D > B$ | | | |
| Excellence Level | $A_1$ | $B_2$ | $C_2$ | $D_3$ |

Note: $X_1$, $X_2$, $X_3$, mean values of the first, second, and third levels of different factors, respectively; $A$, reactive dye (%); $B$, penetrant (%); $C$, $KH_2PO_4$ (%); $D$, pH; $R$, range; $A_1$, reactive dye at a level of 0.2%; $B_2$, penetrant at a level of 0.05%; $C_2$, $KH_2PO_4$ at a level of 0.2%; $D_3$, pH at a level of 5.5.

**Table 6.** Variance analysis of the color difference of dyed red-heart Chinese fir wood before and after soaking in water for 6 h.

| Source of the Variance | Degree of Freedom | Sum Squares of Mean Deviation | Mean Square | *F* | *p* | |
|---|---|---|---|---|---|---|
| Reactive dye (%) | 2 | 65.899 | 32.95 | 10.282 | 0.000 | *** |
| Penetrant (%) | 2 | 0.455 | 0.227 | 0.071 | 0.932 | |
| $KH_2PO_4$ (%) | 2 | 11.985 | 5.992 | 1.87 | 0.160 | |
| pH value | 2 | 4.962 | 2.481 | 0.774 | 0.464 | |
| Error | 18 | 288.421 | 3.205 | | | |
| Total | 26 | 372.509 | | | | |

Note: ***, the significant difference ($p < 0.05$)

### 3.3. Effect of Dye Fluid on Wood Basic Density

The WBD of the experimental and *CK* groups is shown in Figure 3. In the nine groups of dyeing treatment for the orthogonal tests, only treatment No.1 (reactive dye 0.2%, penetrant 0.01%, $KH_2PO_4$ 0.1%, pH value 3.5, WBD 0.327 g/cm$^3$), treatment No.4 (reactive dye 0.5%, penetrant 0.01%, $KH_2PO_4$ 0.2%, pH 5.5, WBD 0.316 g/cm$^3$) and No.6 (reactive dye 0.5%, penetrant 0.1%, $KH_2PO_4$ 0.1%, pH 4.5, WBD 0.312 g/cm$^3$) had significant effects. These three groups were significantly higher than that of the *CK* group, which was 19.4%, 15.2%, 9.3%, and 9.1% higher than that of the *CK* group. There was no significant difference in the basic density between the experimental and *CK* groups. The average WBD in the experimental groups was 5.9% higher than that of the *CK* group after one year.

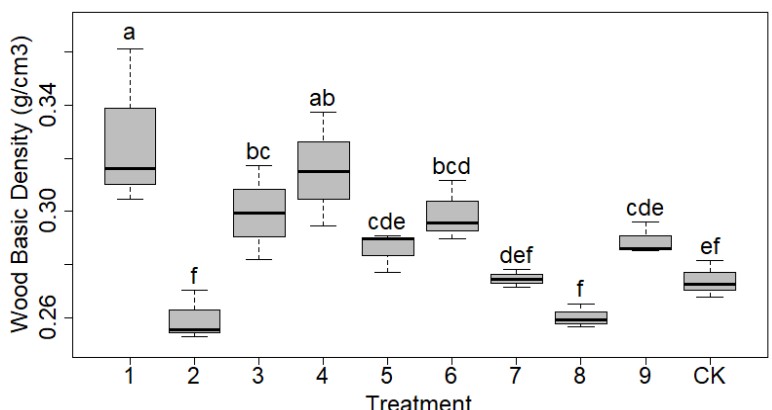

**Figure 3.** Variance analysis of the wood basic density of dyed red-heart Chinese fir with different dyeing treatments. The same lowercase letters indicate no significant difference.

Through the range differential analysis (Table 7), we found that the range of the WBD of the *A*, *B*, *C*, and *D* factors was 0.025, 0.037, 0.009, and 0.023, respectively. That is, the order of the influence effect of the four factors on the WBD of dyed tree wood was penetrant % (*B*) > reactive dye % (*A*) > pH (*D*) > $KH_2PO_4$% (*C*).

**Table 7.** Range analysis of the wood basic density of dyed red-heart Chinese fir wood.

| Horizontal Analysis | Factors | | | |
|---|---|---|---|---|
| | *A* | *B* | *C* | *D* |
| $X_1$ | 0.296 | 0.306 | 0.296 | 0.301 |
| $X_2$ | 0.300 | 0.269 | 0.288 | 0.278 |
| $X_3$ | 0.275 | 0.296 | 0.287 | 0.292 |
| *R* | 0.025 | 0.037 | 0.009 | 0.023 |
| Sort Factors | *B* > *A* > *D* > *C* | | | |

Note: $X_1$, $X_2$, $X_3$, mean values of the first, second, and third levels of different factors, respectively; *A*, reactive dye (%); *B*, penetrant (%); *C*, $KH_2PO_4$ (%); *D*, pH; *R*, range.

According to Figure 4, the basic density of dyed red-heart Chinese fir wood decreased with the mass fraction of reactive dye, and the basic density of all three concentrations was higher than that of the mass fraction of penetrant and the pH value, but decreased with the increase in the mass fraction of $KH_2PO_4$. The basic density of dyed red-heart Chinese fir trunks was used as the evaluation index. The best dyeing process for red-heart Chinese fir live dyeing was $A_2B_1C_1D_1$, which meant the best formula was: mass fraction of reactive dye of 0.5%, mass fraction of penetrant of 0.01%, mass fraction of $KH_2PO_4$ of 0.1%, and pH value of dyeing solution of 3.5.

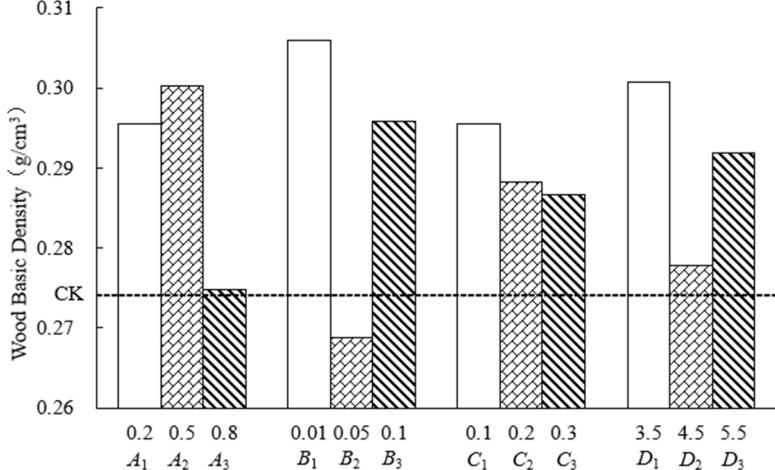

**Figure 4.** Effects of four factors (all with three levels) on wood basic densities of dyed red-heart Chinese fir wood after one year. $A$ is reactive dye (%) with three levels of 0.2% ($A_1$), 0.5% ($A_2$), and 0.8% ($A_3$). $B$ is penetrant (%) with three levels of 0.01% ($B_1$), 0.05% ($B_2$), and 0.10% ($B_3$). $C$ is $KH_2PO_4$ (%) with three levels of 0.1% ($C_1$), 0.2% ($C_2$), and 0.3% ($C_3$). $D$ is pH with three levels of 3.5 ($D_1$), 4.5 ($D_2$), and 5.5 ($D_3$). $A_2B_1C_1D_1$ was the best formula.

### 3.4. Comprehensive Evaluation of the Dyeing Process

Table 8 lists the best process of dyeing and the evaluation index by color difference, colorfastness, and basic density. The results showed that the best level combination was different under each evaluation index.

**Table 8.** The best combination of dyeing solutions under different evaluation indexes and the order of four factors.

| Evaluating Indicator | Optimum Combination | | | | Sort Factors |
|---|---|---|---|---|---|
| | Reactive Dye (%) | Penetrant (%) | $KH_2PO_4$ (%) | pH | |
| Wood color difference | $A_3$ (0.8) | $B_2$ (0.05) | $C_3$ (0.3) | $D_1$ (3.5) | $A > D > B > C$ |
| Wood colorfastness | $A_1$ (0.2) | $B_3$ (0.10) | $C_2$ (0.2) | $D_3$ (5.5) | $A > C > D > B$ |
| Wood basic density | $A_2$ (0.5) | $B_1$ (0.01) | $C_1$ (0.1) | $D_1$ (3.5) | $B > A > D > C$ |

Note: $A$ is reactive dye (%) with three levels of 0.2% ($A_1$), 0.5% ($A_2$), and 0.8% ($A_3$). $B$ is penetrant (%) with three levels of 0.01% ($B_1$), 0.05% ($B_2$), and 0.10% ($B_3$). $C$ is $KH_2PO_4$ (%) with three levels of 0.1% ($C_1$), 0.2% ($C_2$), and 0.3% ($C_3$). $D$ is pH with three levels of 3.5 ($D_1$), 4.5 ($D_2$), and 5.5 ($D_3$).

To further evaluate the best dyeing process of red-heart Chinese fir live dyeing, according to the formula, the evaluation matrices $RA$, $RB$, $RC$, and $RD$ of four influencing factors including the mass fraction of reactive dye, the mass fraction of penetrant, the mass fraction of $KH_2PO_4$, and the pH value, the weight of each index was obtained, respectively: the weight of basic density was 0.3048, the weight of total color difference was 0.5210, and the weight of total colorfastness was 0.1742.

According to the weight of each evaluation index, the largest weight was $\Delta E_1^*$. If only the dyeing effect was considered, the best dyeing process for red-heart Chinese fir live dyeing was $A_3B_2C_3D_1$. The mass fraction of reactive dye was 0.8%, the mass fraction of penetrant was 0.05%, the mass fraction of $KH_2PO_4$ was 0.3%, and the pH value of the

dyeing solution was 3.5. We combined all the indicators and normalized them to obtain *BA, BB, BC, BD*:

$$BA = (0.3115, 0.3152, 0.3733)$$
$$BB = (0.3270, 0.3452, 0.3278)$$
$$BC = (0.3318, 0.3338, 0.3344)$$
$$BD = (0.3601, 0.3113, 0.3582)$$

In the *BA* group, since $BA_3$ (0.3733) > $BA_2$ (0.3152) > $BA_1$ (0.3115), the mass fraction of reactive dye selected was the third level (0.8%); the mass fraction of the penetrant took the second level (0.05%), the mass fraction of $KH_2PO_4$ was 0.3%, and the pH value was the first level (3.5). Based on a fuzzy comprehensive evaluation, three indexes, including the color difference ($\Delta E_1^*$), basic density, and the colorfastness difference ($\Delta E_2^*$), the best dyeing process combination for red-heart Chinese fir live dyeing was $A_3B_2C_3D_1$: the mass fraction of reactive dye was 0.8%, the mass fraction of penetrant was 0.05%, $KH_2PO4$ was 0.3% and the pH value was 3.5.

## 4. Discussion

### 4.1. Effects of Dyeing Factors on the Color Difference

In this study, the variation range of lightness ($L^*$) of dyed wood was 63.17~73.57 NBS, the variation range of the red-green axis color index ($a^*$) was 13.73~26.94 NBS, and the variation range of the yellow-blue axis color index ($b^*$) was 11.04~18.67 NBS. Compared with the range of lightness ($L^*$) values (63.11~82.87 NBS), the range of the red-green axis color index ($a^*$) (6.38~13.10 NBS), and the range of the yellow-blue axis color index ($b^*$) (14.18~24.30 NBS), which were measured by Chen [49] et al., there was little difference in lightness. The color index of the red-green axis was much greater than that of natural heartwood, and the value of the yellow-blue axis was slightly lower, which indicated that the dyed red-heart Chinese fir was darker than that of the natural heartwood, and the dyeing effect on red-heart Chinese fir live wood was better. According to the test results, the reactive dye had a significant influence on the total color difference ($\Delta E_1^*$) of the dyeing material, and it also increased with increasing reactive dye %($\Delta E_1^*$). The results were the same as those obtained by Xing et al. [50] and Li et al. [51] for dyeing poplar and basswood with acid dyes and reactive dyes.

This was mainly because after dye injection, the red-heart Chinese fir depended on its tracheid to carry out vertical infiltration into the equiaxial tissues, which accounted for more than 90% of its total volume [52–54], and the infiltration path was small and there were many obstacles [55]. With the increase in reactive dye content injected into red-heart Chinese fir, the probability of dye breaking through the intermolecular coulombic force and contacting the wood surface fiber increased under the action of a penetrating agent [18,56,57]. In addition, penetrants can also improve the activity of polar molecules in the internal tissue of wood and weaken the selectivity of the internal tissue of wood, resulting in an increase in dye molecules in wood and wood adsorption, increasing the color difference of wood [58,59].

### 4.2. Effect of the Dyeing Factors on Colorfastness

Colorfastness is a measure of the strength of dye molecules to bind with wood. It is an important index to test the colorfastness of dyed wood and an important index to measure the dyeing performance of wood. In our research, the color difference of dyed wood before and after soaking in water was 2.30~5.12 NBS, which belonged to the range of perceptible and recognizable by the human eye. Hu et al. [60] used three kinds of reactive dyes to dye cellulose and lignin from common tree species in Fujian and found the same result. They found that when the wood was stained with reactive dyes, reactive groups in the dye molecules reacted with the wood to form stable chemical bonds. Reactive dyes are considered one of the most stable dyes [61]. The dye molecules can penetrate the intercellular layer through holes in the cell wall of wood, form covalent bonds with the

active groups contained, and become a part of the fiber molecules; thus, the dyeing material has a high water resistance [62,63].

Gu et al. [64] and Gao [65], after the same experiment, found that dye pH affected the dispersion degree of acid dyes in water, and the combination of wood and speed, when the pH of dye solution was small, short of wood dyeing time and dye distribution uniformity and stability in the wood, and before and after dyeing color difference, was small, as low pH can cause corrosion of wood. The pH of the dye was high and the dyeing time was relatively long. It could be seen that pH influenced colorfastness.

### 4.3. Effect of the Dyeing Fluid on WBD

After 1 year of dyeing treatment, the average basic density of the nine experimental groups was 5.9% higher than that of the *CK* group. The penetrant had the most significant effect on the basic density of dyed fir wood, because the penetrant was a nonionic surface reactive agent that reduced the surface tension of the dye solution and increased the wettability of the dye solution at the wood interface. This was the same as the research results of the rapidly growing Chinese fir dyeing process by single-factor and multifactor orthogonal tests with acidic dyes by Ma et al. [59], who found that the penetrant could increase the permeability of the dye. Moreover, penetrants can improve the activity of the internal tissues of polar molecules and reduce the absorptive capacity of the internal tissues of wood, which can significantly improve the ability of water and dye in wood impregnation and promote more dye in the wood-dyeing fluid [66].

$KH_2PO_4$ was added to the dye solution to alleviate the problem of phosphorus deficiency during the growth of red-heart Chinese fir. In the experiment, the basic density of dyeing red-heart Chinese fir was considered as the evaluation index. $KH_2PO_4$ had a slight effect on the basic density of dyed red-heart Chinese fir, which may have been due to the influence of other factors, resulting in the effect of dyeing solution $KH_2PO_4$ not being significant.

### 5. Conclusions

Based on the results, we compared the color difference ($\Delta E_1*$) and colorfastness ($\Delta E_2*$) before and after the different dyeing treatments, and there was a significant improvement. After dyeing for 1 year, the effects of different treatments on the basic density of wood was also significantly improved. According to the fuzzy comprehensive evaluation of these three indicators, the best dyeing process combination for red-heart Chinese fir live dyeing was: reactive dye of 0.8%, penetrant of 0.05%, $KH_2PO_4$ of 0.3%, and pH value of 3.5. Through studying the perforation dyeing technology on living trees, the results will play a theoretical guiding role in the fast-cultivated, high-value decoration of red-heart Chinese fir and the cultivation of varieties.

**Author Contributions:** Conception and design of experiments, X.D.; methodology and performance of experiments, R.Q. and Y.W.; software and statistical data analyses, X.D., Z.H., and R.Q.; writing—original draft, X.D., R.Q. and Y.W.; data curation and validation, X.D., Z.H., Y.W., S.O. and W.X.; project administration, X.D. All authors have read and agreed to the published version of the manuscript.

**Funding:** This work was funded by the National Key Research and Development Program of China (2016YFD0600303).

**Institutional Review Board Statement:** Not applicable.

**Informed Consent Statement:** Not applicable.

**Data Availability Statement:** Not applicable.

**Acknowledgments:** The authors thank the Chenshan Forestry Farm of Anfu County, Jiangxi Province for supporting the fieldwork.

**Conflicts of Interest:** The authors declare no conflict of interest.

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
