# Peer review of "Effect of Perforation Dyeing Technique on Color Difference, Colorfastness, and Basic Density of Living Red-Heart Chinese Fir"

_forests, doi:10.3390/f12121721_

Round 1

Reviewer 1 Report

This manuscript shows that the authors did decent research. However, addressing the following comments can improve the quality of the paper:

  1. Add more recent citations and compare your results with theirs in "Discussion".
  2. Explain what basic density is and how it is different than the regular density
  3. Elaborate on the orthogonal test (L9(3)4).

Author Response

Point 1: Add more recent citations and compare your results with theirs in "Discussion".

Response 1: Thank you very much for your comments on our manuscript. We have  added references [50], [51], [61], [62], [65], [66], [67] in "Discussion", and compared and analyzed the results with them.

Point 2: Explain what basic density is and how it is different than the regular density

Response 2: Thanks. We have added explanations in lines 177-184.

“Wood regular density is divided into four kinds. And the wood basic density (WBD) and wood air-dry density are commonly used. WBD is one of the most crucial wood properties in the tree and mainly determined the end-use of wood for the industry. Wood air-dry density is suitable for measuring the density of wood in dry air for a long time [40]. The test object is Chinese fir in the natural environment, so it is more suitable for measuring the WBD. The WBD of samples was determined by following per under GB/T 1933-2009 [41], and 3 replicates were taken for each group of treated samples.”

Point 3: Elaborate on the orthogonal test (L9(3)4).

Response 3: Thanks. The elaboration of orthogonal experiments has been supplemented in lines 128-133:

We established the L9(3)4 Orthogonal experimental, which included 4 influencing factors all with 3 concentration levels. In the experiment, there were 9 treatments with 3 replicates per treatment, and the control group used distilled water. According to the orthogonal table, the experimental treatment can be distributed evenly, and the number of experiments can be reduced.

Reviewer 2 Report

This study looks at a method(s?) to induce red staining in Chinese fir heartwood. Unfortunately, key methods and background are missing so it is very hard to interpret the results and discussion. Explanations, background, and methodology need to be added to the introduction of common dye techniques, and specific methodology for this test needs to be added to the methods (see below for specifics).

This paper has potential, but right now the missing information makes it hard to understand what was done, and what the impact of the results might be.

  • lines 41-52 are confusing. This is the introduction and literature review, so I am unsure why methodology is being discussed
  • the introduction as a whole does not give me a clear picture of wood dyeing. Many words are used that I am not familiar with, and I dye wood for a living. I suggest adding to the introduction some sort of wood dyeing flow chart, that discusses popular methodology, then moves into whatever type of dyeing is being discussed here, with a quick note of what that process entails. Is the wood being pressure treated with a chemical? Dipped? Is the living tree being watered with aniline dyes? Is the soil of the living tree being pH treated? What is the process we are actually talking about, what are the other processes for the same effect, and what are the pros and cons of all of these processes?
  • lines 92-99: here is where I should get a clear idea of how this dyeing is taking place, but I don't. Are live trees being fed phosphorus? 
  • the methods lack the actual dyeing method and discussion of how this pH change came about. It is unclear if living trees are being dyed, or the trees are cut and then somehow dyed.
  • Table 3 is the first time the kinds of dyes are noted. This needs to be in the methods and very well spelled out
  • line 345 notes 'dye injection.' Where was the dye injected? How? Into living or dead trees? How did it get taken up into the heartwood and not the much more permeable sapwood?
  • Was color fade monitored during this one year study? Many dyes are not color stable over the long term. What were the CIE Lab numbers after a year, compared to the originals? What is the delta E?

Author Response

Response 1: We highly appreciate the valuable comments and suggestions. We have revised the manuscript based on the reviewer’s comments and attempted to address each point.

  • We modified this part. Lines 41-45 are an introduction to wood dyeing technology. This part mainly introduces the role and value of wood dyeing technology. The clear process of wood dyeing was supplemented in3. Dyeing method.
  • We have revised the description in lines 52-66 and supplemented other dyeing methods and their dyeing processes and characteristics. The flow chartof various dyeing methods has not been found, so the different dyeing methods are described in chronological order in lines 67-84.
  • In lines 52-66 we analyzed the advantages and disadvantages of the different processes.

By comparing the advantages and disadvantages of different methods, we finally decided to use the standing wood dyeing method. The action principle of the standing wood dyeing method has been added in lines 63-66: “Standing wood dyeing is to inject the dye solution from the base of the trunk into the growing living tree, or just cut trees still have vitality so that the dye solution is transported to various parts through the capillary of wood, which is suitable for small wood with more sapwood [18,19].” And the specific operation methods, experimental procedures, and wood treatment methods have been supplemented in 2.3. Dyeing method.

Response 2: Thanks for the positive comment

  • The actual dyeing method and specific test process have been added in 2.3. Dyeing method. We also added the discussion of how this pH change came about in lines 120-125, “the pH of dyeing solution will affect the time and color difference of wood dyeing [35]”
  • We used the dyeing of living trees. And we added the specified dyeing methods and processes in3. Dyeing method.In lines109-127, The selection reasons and functions of dyes (reactive dye, penetrant, KH2PO4, and pH), which in Table 3 are supplemented. The addition method of KH2PO4 has been explained in 116-119, “we added different concentrations of KH2PO4 in the dyes.” 

Response 3: Thanks. This makes my content clearer and makes it easier for the reader to read.

  • In lines 145-148, we have added the experimental process, “when injecting, avoid the damaged (moth-eaten, disaster, etc.) parts of the trunk, and drill holes with a charging hand drill 15 cm from the ground in the south and north directions of the trunk. The hole diameter is about 0.5 cm, and the hole is deep to the middle of the trunk”.
  • In lines 150-152, there's no color fade monitored, because, after one year of the experiment, the color difference, colorfastness, basic density were determined by cutting the sample wood.The best dyeing process can be obtained by these three indexes.
  • In lines 155-162, the CIE Lab numbers in Figure 2 are the changes of color difference after one year. ΔE* is the total color difference.

Round 2

Reviewer 2 Report

These edits address my questions.